# Clinical Diagnosis of Gastrointestinal Stromal Tumor (GIST): From the Molecular Genetic Point of View

**DOI:** 10.3390/cancers11050679

**Published:** 2019-05-16

**Authors:** Chiao-En Wu, Chin-Yuan Tzen, Shang-Yu Wang, Chun-Nan Yeh

**Affiliations:** 1GIST Team, Division of Hematology-Oncology, Department of Internal Medicine, Chang Gung Memorial Hospital, Linkou branch, Chang Gung University, Taoyuan 333, Taiwan; jiaoen@gmail.com; 2Forlab Clinic, F2, No 14, Sec 2, Zhongxiao East Rd, Taipei 100, Taiwan; visitor@clinic4lab.com; 3GIST Team, Department of Surgery, Chang Gung Memorial Hospital, Chang Gung University, Taoyuan 333, Taiwan; shangyuwang@gmail.com

**Keywords:** diagnosis, gastrointestinal stromal tumor, GIST, molecular genetic

## Abstract

Gastrointestinal stromal tumors (GISTs) originating from the interstitial cells of Cajal are mesenchymal tumors of the gastrointestinal tract and have been found to harbor *c-KIT* mutations and KIT (CD117) expression since 1998. Later, *PDGFRA* mutations, *SDH* alterations, and other drive mutations were identified in GISTs. In addition, more and more protein markers such as DOG1, PKCθ were found to be expressed in GISTs which might help clinicians diagnose CD117-negative GISTs. Therefore, we plan to comprehensively review the molecular markers and genetics of GISTs and provide clinicians useful information in diagnostic and therapeutic strategies of GISTs. Twenty years after the discovery of KIT in GISTs, the diagnosis of GISTs became much more accurate by using immunohistochemical (IHC) panel (CD117/DOG1) and molecular analysis (*KIT/PDGFRA*), both of which constitute the gold standard of diagnosis in GISTs. The accurately molecular diagnosis of GISTs guides clinicians to precision medicine and provides optimal treatment for the patients with GISTs. Successful treatment in GISTs prolongs the survival of GIST patients and causes GISTs to become a chronic disease. In the future, the development of effective treatment for GISTs resistant to imatinib/sunitinib/regorafenib and *KIT/PDGFRA*-WT GISTs will be the challenge for GISTs.

## 1. Introduction of Gastrointestinal Stromal Tumors (GISTs)

Gastrointestinal stromal tumors (GISTs) are a subgroup of mesenchymal tumors arising from the gastrointestinal (GI) tract and were considered smooth muscle tumors (such as leiomyomas or leiomyosarcomas), based on their histologic characteristics. The entity of GIST was not disclosed until the identification of KIT (CD117) expression, *c-KIT* mutations, and the discovery of the origin of GISTs [1,2].

In 1998, Hirota and his colleagues discovered gain-of-function mutations in the intracellular domain of the *c-KIT* proto-oncogene in GISTs as well as near-universal expression of the KIT protein in GISTs by immunohistochemical (IHC) staining [2]. In addition, they proposed that GISTs might originate from the interstitial cells of Cajal (ICCs), the pacemaker cells of the intestines, which express both KIT and CD34 [2], and whose normal development is dependent on the interaction between KIT and its ligand, stem cell factor (SCF) [3,4,5,6,7,8]. At the same time, other research groups confirmed these findings by reporting similar results and that CD117 is a more sensitive and specific marker than CD34 for GISTs [9]. In addition, GISTs show morphological and immunophenotypic similarities (CD117+/CD34+/Vimentin+) to ICCs, suggesting that ICCs are the precursors for GISTs [10,11]. Histologically, not all GISTs are composed of spindle cells, which accounts for only 70% of GISTs, and other subtypes such as epithelioid cells and mixed spindle and epithelioid cells account for 20% and 10% of GISTs. Therefore, the molecular and genetic biomarkers provide additional information for the diagnosis of GISTs.

With the understanding of the molecular biology of GISTs and the discovery of effective targeted therapy against KIT, GISTs became more important than before, and the nature of the disease becomes chronic. Several small molecular compounds that target the KIT protein, such as imatinib [12,13], sunitinib [14], and regorafenib [15], are effective in treating advanced GISTs and have been approved for the treatment of advanced GISTs. All KIT inhibitors are widely used in our routine practice for patients with advanced GISTs and significantly improve the survival of such patients [16,17,18,19]. Therefore, it is crucial to make an accurate diagnosis of GISTs so that optimal treatment can be used for patients with GISTs. Here, we discuss the diagnostic development in GISTs, focusing on protein expression by IHC and genetic alterations (Figure 1).

## 2. The Diagnosis of GIST from the IHC Point of View (>95%)

### 2.1. IHC of CD117

Before the identification of CD117 expression in GISTs, CD34 was considered the best marker for GIST, but it was neither sensitive (only for two-thirds of GISTs) nor specific (immunoreactive in fibroblastic and endothelial cell tumors) [20]. The findings of KIT and ICCs, *c-KIT* proto-oncogene mutations and CD117 expression were identified in most GISTs, which opened a new era of molecular diagnosis in GISTs.

In early studies with a limited number of cases, the positive rate of CD117 expression was 76–100% [2,9,10,21,22,23,24,25]. The largest series of 1168 GIST patients reported CD117 was expressed in 94.7% of 1040 GISTs [26]. The variation among studies in the positive rate of CD117 in GISTs possibly resulted from the distribution of primary locations, different KIT antibodies, and limited numbers in some reports. On average, approximately 95% of GISTs expressed CD117. In addition, the mimics of GISTs, such as leiomyomas, leiomyosarcomas and schwannomas, were near-universally negative for CD117, indicating that CD117 is a highly sensitive and specific marker for GISTs [9,10,20].

For the reason that the majority of GISTs express KIT which was retained in refractory GISTs after small molecule treatment, anti-KIT monoclonal antibodies were investigated in preclinical setting and might move forward to clinical trials in the future [27,28].

### 2.2. IHC of DOG1

Although CD117 is a sensitively and specifically expressed marker in GISTs, CD117 is not universally expressed in all GISTs; however, it can be expressed in other tumors, such as melanomas, adenoid cystic carcinomas, Merkel cell carcinomas, Kaposi sarcomas, liposarcomas or even leiomyosarcomas (rarely) [9,10,20]. Therefore, additional markers are needed to compensate for the weakness of CD117. A novel marker, DOG1 (Discovered On GIST-1), was first discovered in 2004 [29] and is widely used in most pathology laboratories.

*DOG1*, which encodes the chloride channel protein anoctamin 1 (ANO1/TMEM16A), is a novel gene discovered in GISTs. DOG1 is involved in cell proliferation and tumorigenesis in various cancers [30]. Regarding DOG1 in GISTs, one study that investigated the physical importance of ANO1 in ICCs, and showed that the intestinal ICCs expressed ANO1 not only for slow wave generation (regulation of ICC cell excitability and intestinal movement rhythm) but also for ICC cell proliferation [31]. ANO1 knockout mice have fewer proliferating ICCs in the small intestine, and primary culture of ICCs from ANO1 knockout mice showed less proliferation. The pathological role of DOG1 was unclear in GISTs until 2013. Simon and his colleagues found that DOG1 regulated the anti-angiogenic factor IGFBP5, leading to the modulation of IGF/IGFR signaling in the tumor microenvironment, which did not involve KIT expression and KIT-dependent pathways [32]. Other findings were reported that DOG1 exhibited little effect on GIST cell viability and proliferation in vitro [33].

DOG1 is considered a sensitive and specific marker of GISTs regardless of CD117 expression. DOG1 is also independent of KIT or platelet-derived growth factor receptor α (PDGFRA) mutation status in GISTs [29]. In the first report of DOG1 in GISTs, DOG1 was found to be expressed in 136 of 139 (97.8%) scorable GISTs. In addition, in this study, all 8 GISTs with *PDGFRA* mutations were positive for DOG1, while more than half of them failed to express CD117. In 438 non-GIST neoplasms, only 4 (<1%) out of 438 were immunopositive for DOG1, indicating that DOG1 is extremely specific for GIST [29].

In the largest series including 1168 GISTs of different sites and histologic subtypes and 672 other tumors and normal tissues, a total of 986 of the 1040 GISTs (94.8%) from the different sites in the GI tract and abdomen were positive for DOG1 (monoclonal antibody clone K9) [26]. In this report, DOG1 and CD117 expression in the majority of GISTs (960 of 1040, 92.3%) was generally concordant, particularly in gastric spindle GISTs. DOG1 performed slightly better in gastric epithelioid GISTs, including *PDGFRA*-mutant GISTs, but was slightly less sensitive than CD117 in the intestinal GISTs. Although DOG1 was highly specific to GISTs, DOG1 was stained in other mesenchymal tumors, including uterine type retroperitoneal leiomyomas (5/42), peritoneal leiomyomatosis (4/17), and synovial sarcomas (6/37). In addition, although DOG1 was also expressed in esophageal squamous cell carcinomas and gastric carcinomas, those carcinomas can be easily distinguished by histologic morphology.

An important issue should be noted that not all antibodies against DOG1 perform equally in GISTs. A study compared two different anti-DOG1 monoclonal antibodies, DOG1.1 and K9, which were positive in 80.5% and 96.1% of 668 GISTs, respectively. In addition, DOG1.1 and K9 antibodies were positively expressed in 5 (20.0%) and 19 (76.0%) samples in 25 KIT-negative GISTs, respectively [34]. Therefore, obtaining an optimal antibody is important in clinical practice.

In conclusion, DOG1 showed similar and compatible sensitivity and specificity with CD117, and both can compensate for the weakness and limitations in the diagnosis of GIST. The combination of both CD117 and DOG1 in an IHC panel covers more than 98% of GISTs in clinical practice. Although either CD117 or DOG1 can be detected in non-GIST tumors, appropriate clinical and pathological assessments in combination with CD117/DOG1 are important for correctly diagnosing GIST.

### 2.3. IHC of PKCθ and Others

The diagnosis of GISTs without expression of CD117/DOG1 (1–2%) is difficult to render by IHC staining even if they have KIT/PDGFR mutations. Therefore, additional proteins were investigated to support the diagnosis of GISTs, particularly in GISTs with negative expression of CD117/DOG1.

Protein kinase C-θ (PKCθ) is a serine-threonine protein kinase that is expressed in the ICCs in the digestive tracts of guinea pigs [35,36]. Overexpression of the PKCθ gene was found in GISTs [37,38]. PKCθ was considered a useful IHC marker for the diagnosis of GISTs regardless of KIT expression. The studies to evaluate the usefulness of PKCθ expression in GISTs showed that PKCθ is sensitive but less specific with immunoreactivity to other spindle cells, such as schwannomas and smooth muscle tumors [39,40,41,42]. Although PKCθ was expressed in some KIT-negative GISTs, providing supportive evidence for the diagnosis of GISTs, the lower sensitivity of PKCθ in KIT-negative than KIT-positive GISTs and lower specificity of PKCθ than KIT expression limits its usefulness in GISTs [42].

Another protein, nestin, is an intermediate filament protein and is found to be expressed in immature cells, such as neuroectodermal stem cells and skeletal muscle progenitor cells, and tumors originating from these cells [43]. In addition, nestin has been found to regulate mitochondrial dynamics and alter intracellular ROS levels in GISTs, which provides a new mechanism where nestin mediates the proliferation and invasion of GISTs [44]. However, similar to PKCθ, nestin is sensitive but less specific than KIT/DOG-1 [45].

### 2.4. Conclusions of IHC

There are several proteins showing different sensitivities and specificities in the diagnosis of GISTs. CD117 and DOG1 are the most sensitive and specific markers that can cover ~99% of GISTs and are rarely expressed in non-GIST tumors. Although CD34 was widely used in the diagnosis of GISTs before CD117, it is no longer used for this purpose because of its lower sensitivity than CD117. Similarly, the role of PKCθ and nestin in the diagnosis of GISTs is limited because of their lower specificity in other spindle cell tumors, such as schwannomas and smooth muscle tumors. Other markers, such as smooth muscle markers (SMA and desmin) and the neural marker (S100), are occasionally useful when excluding the diagnosis of GISTs and are needed for the diagnosis of other gastrointestinal mesenchymal tumors, such as schwannomas (S100) and smooth muscle tumors (SMA and desmin) [45] (Figure 2).

## 3. The Diagnostic View of Genetic Alterations in GISTs

### 3.1. KIT Mutations (80–85%)

Approximately 80% of GISTs have KIT gene mutations, leading to constitutive activation of the KIT receptor and downstream signaling pathways that stimulate cell survival, growth, and proliferation [2,25,46,47]. These “gain-of-function” mutations in the KIT gene in GISTs were identified in different exons of the gene and include point mutations, deletions, or insertions, all of which can occur in both sporadic and hereditary cases. Therefore, the KIT gene has been considered a primary factor in the tumorigenesis of GISTs [2,25,46,47,48].

There is no mutational hotspot of KIT mutations in GISTs, but some exons are affected more often than others. The majority of KIT mutations in GISTs (~70%) affect exon 11, which encodes the intracellular juxtamembrane domain of the receptor [2,47]. This region usually has an autoinhibitory function on kinase activation, which is alleviated by the mutation. Mutations in exon 9 affecting the extracellular ligand-binding domain are detected in 12–15% of cases. Primary mutations in the kinase domain (exon 13, ATP-binding; exon 17, activation loop) are rare [47]. Importantly, however, these mutations are seen at a high frequency as secondary mutations in imatinib-resistant GISTs [47,49]. The KIT mutations in different regions can affect the response to targeted therapy, providing guidance for choosing an appropriate agent with optimal dosage. For example, imatinib works better in tumors with mutations in exon 11 mutations than those with exon 9 mutations [50,51]. In terms of a second-line setting, sunitinib works better than imatinib escalation in tumors with non-exon 9 mutations [17], whereas imatinib escalation shows better activity in tumors with exon 9 mutation than those with exon 11 mutations [52]. In addition, secondary KIT exon 17 mutations contributed to 30–40% of KIT secondary mutations, which accounts for the resistance to imatinib or sunitinib in GIST patients [53,54]. In this situation, regorafenib shows therapeutic activity in these patients [49].

### 3.2. PDGFRA Mutations (5–7%)

Approximately 15–20% of GISTs lack *KIT* mutations and are so-called *KIT* wild-type (*KIT-*WT) GISTs. In 2003, Heinrich et al. investigated possible alternative receptor tyrosine kinase (RTK) oncoproteins using immunoprecipitations with polyclonal panRTK antisera and found that phospho-PDGFRA was the predominant phospho-RTK in GIST478, a *KIT*-WT GIST cell line. In this regard, 35% (14 of 40) of *KIT*-WT GISTs harbored activated PDGFRA mutations [55]. Furthermore, phospho-PDGFRA expression was restricted to *KIT*-WT GISTs that expressed low to undetectable KIT, indicating that phospho-KIT and phospho-PDGFRA expression was mutually exclusive in GISTs. GISTs expressing either KIT or PDGFRA oncoproteins have similar mechanisms of tumorigenesis and progression via oncoprotein-driven signal transduction. Therefore, *KIT* and *PDGFRA* mutations are alternative and mutually exclusive oncogenic mechanisms in GISTs [55].

A large-scale study reported that 7.2% (80 of 1105) GISTs harbor *PDGFRA* mutations [56], which was higher than the previously reported frequency. This is most likely because of the referral of *KIT*-negative cases. *PDGFRA*-mutant GISTs preferentially occurs in epitheliolid morphology and arises exclusively in stomach while most spindle GISTs dominate most *KIT*-mutant GISTs and could arise at any sites of the GI tract [56].

Similar to *KIT* mutations in GISTs, there is no single hotspot for *PDGFRA* mutations in GISTs. The *PDGFRA* mutations occurred in exons 12, 14 and 18, and D842V in exon 18 was the most frequent mutation (62.6%) in a collection of 289 GISTs with *PDGFRA* mutations [56]. Not all activating mutations in *PDGFRA* are biologically equivalent. Most of the mutations in exon 18, particularly D842V [55,56,57], were characterized by their relative insensitivity to imatinib, with the exception of D842Y, which is sensitive to imatinib but confers resistance to sunitinib, as shown in in vitro experiments [52]. Nevertheless, most GISTs with *PDGFRA* mutants other than the D842V substitution are still responsive to imatinib, so mutation screening is important and helpful in the management of GISTs [58]. In Asia, the *PDGFRA* mutation rate was less than 5%, which is lower than in Western countries [50,59,60].

*PDGFRA-*mutant GISTs express PDGFRA rather than KIT, from which arose the research of monoclonal antibody against PDGFRA to treat the patients with *PDGFRA-*mutant GISTs. A phase II study of olaratumab demonstrated the disease control in *PDGFRA-*mutant GISTs but not *PDGFRA-*WT GISTs [61].

In conclusion, a subset of GISTs lacking *KIT* mutations has activating mutations in the related RTK PDGFRA [55,57,62]. Because of this variability, the assessment of mutational status is important for decision making and treatment purposes in clinical practice for either advanced disease or for the adjuvant/neoadjuvant setting.

### 3.3. KIT/PDGFRA “Wild-Type (WT)” GISTs (5–10%)

Although most GISTs have either a mutation of *KIT* or *PDGFRA* kinases leading to constitutive activation, approximately 10 to 15% of GISTs do not have a detectable *KIT* or *PDGFRA* mutation. They are collectively grouped as *KIT*/*PDGFRA*-WT GISTs. Succinate dehydrogenase (SDH) (also called mitochondria complex II) alteration and loss of SDHB expression were found in a majority of *KIT*/*PDGFRA*-WT GIST patients in a comprehensive cohort study examining molecular subtypes of *KIT*/*PDGFRA*-WT GISTs. Among 84 patients with adequate tissue for analysis, three distinct molecular subtypes were defined, including *SDHX* mutations (66%), *SDHC* promoter hypermethylation (22%) and *SDH* competence (12%) [63]. The former two subtypes are associated with SDH deficiency, and the third subtype can be further divided into *NF1* mutations, *BRAF* V600E mutations and others. The details of such *KIT*/*PDGFRA*-WT GISTs will be discussed later. The genetic classification is summarized in Figure 3.

## 4. CD117/DOG1-Positive GISTs with *KIT*/*PDGFRA* Mutations (80–85%)

Almost all GISTs overexpress CD117/DOG1 (>95%) [26], and approximately 80% and 5% of GISTs have *KIT* or *PDGFRA* gene mutations, respectively, that lead to constitutive activation of RTK. There should be no difficulty diagnosing GIST in this situation. The relationship between *KIT* gene mutations and KIT protein (CD117) expression in GISTs is not entirely concordant. Regarding gene alterations, most *KIT*-mutant GISTs express CD117/DOG1 with exceptions of transcriptional silencing/translational defects in few cases of GISTs with *KIT* mutations. In contrast, *PDGFRA*-mutant GISTs nearly universally express DOG1 [29] but not CD117 [55]. In daily practice, the diagnosis of GIST is mainly based on IHC markers, including CD117 and DOG1, and with the help of genetic analysis.

## 5. CD117/DOG1-Negative GISTs with *KIT*/*PDGFRA* Mutations (<5%)

Although using IHC with CD117/DOG1 provides a simple and clear diagnostic marker for GISTs, immunohistochemically CD117/DOG1-negative GISTs account for less than 5% of GISTs [26,34,64]. In a report enrolling 1040 GISTs, only 27 GISTs (2.6%) were immunoreactively negative for both CD117 and DOG1 [26]. In another study, only 0.9% of GISTs were negative for KIT and DOG1, which were detected by two antibodies against DOG1.1 and K9 [32]. In patients with unclear diagnoses, genetic analysis is necessary to confirm the diagnosis of GIST. These tumors should have either *KIT* or *PDGFRA* mutations; otherwise, the diagnosis of GIST is questionable [65]. Possible mechanisms are available to explain the discordance between gene mutations and protein expression.

(1) When GISTs without CD117 expression were found, *PDGFRA* mutations should be taken into consideration first [55]. GISTs with *PDGFRA* mutations express PDGFRA but have low to undetectable CD117, suggesting that the activation of another oncoprotein in *KIT*-WT GISTs might be associated with *KIT* transcriptional downregulation [55]. Therefore, CD117 and PDFGRA were found to be mutually exclusively expressed in GISTs. However, other sensitive markers, DOG1 and PKCθ, remain expressed in GISTs with *PDGFRA* mutations and compensate for the weakness of CD117, as the expression of both proteins in GISTs is independent of *KIT*/*PDGFRA* mutational status [29,41,65,66].

(2) For CD117-negative, *KIT-*mutated GISTs, it is possible that transcriptional silencing/translational defects occur in an uncommon subpopulation of GISTs via a different mechanism [67]. GISTs with this phenotype appear to be resistant to imatinib therapy because there is no expressed CD117 protein to be targeted by imatinib [65].

## 6. CD117/DOG1-Positive GISTs without *KIT*/*PDGFRA* Mutations (*KIT*/*PDGFRA*-WT) (5–10%)

Some GISTs express CD117/DOG1 but lack *KIT*/*PDGFRA* mutations. These *KIT*/*PDGFRA*-WT GISTs account for 5–10% of all GISTs and are particularly found in pediatric tumors [68] and those arising in patients with neurofibromatosis type 1 (NF1) [69,70,71]. The mechanism of CD117 overexpression in cases without an identifiable mutation in *KIT* is unclear. These GISTs have a poor response to imatinib, possibly related to a lack of driver mutations of *KIT/PDFRA* [70].

GISTs rarely occur in children and young adults, but such GISTs have distinct clinical, molecular and pathologic features in comparison with adult GISTs. Generally, pediatric GISTs have a female predilection, common presentation of chronic gastrointestinal bleeding, more epithelioid than spindle cytomorphology, multifocality in gastric cases, and a trend with regional and distant metastases but a more indolent disease course than adult GISTs [72,73,74,75,76]. Most pediatric GISTs are SDH-deficient and tend to arise within defined syndromes (Carney triad, Carney-Stratakis syndrome), which are discussed below [77,78].

### 6.1. CD117/DOG1-Positive and KIT/PDGFRA-WT GISTs: SDH-Deficient GISTs (>80% of KIT/PDGFRA-WT GISTs)

SDH-deficient GISTs account for 5% to 7.5% of all unselected GISTs [79,80] and the majority (>80%) of *KIT*/*PDGFRA*-WT GISTs. These tumors have a tendency to occur in young females (<18 years) and are restricted to gastric GISTs [63].

The SDH protein complex, also called mitochondrial complex II, is composed of four subunits, in which SDHA and SDHB subunits harbor the main catalytic domain of SDH, while SDHC and SDHD are responsible for the anchoring component which anchors to the inner membrane of mitochondria [81]. Therefore, lacking any subunits of SDH could lead to the instability of the SDH complex resulting in SDHB degradation. In addition, methylation of the SDHC promoter leads to silencing of SDHC expression [82,83]. Therefore, loss of SDHB expression is limited in *KIT*/*PDGFRA*-WT GISTs specifically in SDH-deficient GISTs, whereas *KIT*/*PDGFRA*-mutant GISTs contain intact SDHB expression. IHC staining of SDHB becomes a diagnostic marker for mutations of any of SDH subunits.

SDH-deficient GISTs could result from germline mutations [63]. As mentioned above, most pediatric GISTs and GISTs in patients with Carney triad or Carney-Stratakis syndrome are SDHB-deficient.

Carney triad, a rare nonhereditary syndrome characterized by GISTs, pulmonary chondromas and functional extra-adrenal paragangliomas, was first reported in 1977 by Carney J.A. et al. [84]. In 2002, Carney and Stratakis first reported a new hereditary syndrome, i.e., Carney-Stratakis syndrome, characterized by the combination of familial paraganglioma and gastric GISTs without associated pulmonary chondroma [85]. Some studies [79,86] have demonstrated the role of SDH abnormalities in the Carney triad, as indicated by the loss of SDHB staining in tumors. These findings were further supported by recent studies [82,87,88] that provided evidence of the rare role of germline SDH gene mutations (10%, 6/63) in Carney triad [87], but more frequently, SDHC hypermethylation occurs in Carney triad (sporadic, (90%)).

*KIT*/*PDGFRA*-WT GISTs are more resistant to imatinib but are more sensitive to sunitinib compared with GISTs with *KIT*/*PDGFRA* mutations. In addition, insulin-like growth factor 1 receptor (IGF1R) overexpression was found frequently (71/80, 88.8%) in SDH-deficient GISTs (SDHB-negative GISTs) but rarely (9/625, 1.4%) in SDHB-positive gastric GISTs, suggesting that the loss of SDH is associated with the overexpression of IGF1R, which might involve the oncogenic mechanism of SDH-deficient GISTs and could also be a therapeutic target in GISTs [89]. SDH-deficient GISTs are a unique group of GISTs with an energy metabolism defect, as the key process of tumorigenesis appears to be mediated via overexpression of hypoxia-inducible factors (HIF) proteins and the IGF1R-dependent pathway [90,91].

In clinical practice, IHC staining for SDHB is mandatory when no *KIT* or *PDGFRA* mutations are identified. The absence of SDHB indicates SDH deficiency, and genetic testing for germline abnormalities in the *SDHX* genes is appropriate. IHC of IGF1R provides supportive evidence in diagnosis and possible therapeutic options.

### 6.2. CD117/DOG1-Positive and KIT/PDGFRA-WT GISTs: SDH-Competent GISTs (<20% of KIT/PDGFRA-WT GISTs)

*KIT*/*PDGFRA*-WT GISTs, which are SDH-competent, retain SDH expression and normal methylation. The demographic features of such GISTs are similar to *KIT*/*PDGFRA*-mutant GISTs, which occur in adults and have histology of spindle cells [63]. However, 9/11 (82%) of SDH-competent GISTs occur in the small bowel, which is higher than *KIT*/*PDGFRA*-mutant GISTs. This subgroup of GISTs harbors other genetic abnormalities, some of which have therapeutic implications. Three patients had *BRAF* mutations, and another 3 had *neurofibromin 1 (NF1)* mutations in 11 patients with SDH-competent GISTs [63]. For example, a *BRAF* V600E mutation can be detected in up to 13% of *KIT*/*PDGFRA* wild-type GISTs [92,93], and these *BRAF*-mutant GISTs may be candidates for treatment with BRAF inhibitors [94].

### 6.3. CD117/DOG1-Positive and KIT/PDGFRA-WT GISTs: GISTs in Neurofibromatosis Type I (NF1)

GISTs in NF1 patients have a different pathogenesis than sporadic GISTs. Most of these tumors occur in the small bowel with multifocal tumors [69]. Although all the reported cases in two large series expressed CD117 [69,70], the patients with metastatic tumors have a poor response to imatinib [70]. NF-1-associated GISTs are usually *KIT/PDGFRA*-WT [69], although sporadic *KIT/PDGFRA* mutations are reported in some cases [70].

NF1-associated GISTs exhibit increased signaling through the mitogen-activated protein kinase (MAPK) signaling cascade [95], raising the possibility that treatment with MEK inhibitors could be promising. Finally, comprehensive genomic profiling studies identified a number of gene fusions in *KIT*/*PDGFRA* wild-type GISTs that involve neurotrophic tyrosine kinase receptor type 3 (NTRK3) and fibroblast growth factor receptor 1 (FGFR1), some of which might represent “actionable alterations” [96].

## 7. CD117/DOG1-Negative GISTs without *KIT*/*PDGFRA* Mutations (*KIT*/*PDGFRA*-WT) (<1%)

The question of whether there are *KIT*/*PDGFRA*-WT GISTs without CD117/DOG1-expression is still unknown. However, the answer is debatable and probably yes. How do we make a confident diagnosis via current understandings of molecular biology in GISTs if both IHC and genetic tests are negative to support the diagnosis of GISTs?

In one early report of 1040 GISTs, only 27 (2.6%) of GISTs were negative for both CD117/DOG1, and 8 of 12 CD117/DOG1 negative GISTs were *KIT*/*PDGFRA*-WT GISTs [26]. In another report, there were no GISTs with negative CD117/DOG1 and *KIT*/*PDGFRA*-WT in 99 GISTs [64]. Therefore, the possibility of such a phenotype is quite low, with the limitations of the sensitivity and specificity of the diagnostic tools. The diagnosis of GISTs with negative CD117/DOG1 expression and *KIT*/*PDGFRA* mutations is challenging and should be carefully discussed by the multidisciplinary team [18].

## 8. Germline Mutations vs. Sporadic Mutations

Although the majority of GISTs appear to be sporadic, some patients have one of several familial autosomal dominant syndromes, including NF1, Carney-Stratakis syndrome, Carney triad (10%, mostly sporadic) and primary familial GIST syndrome. All of them other than primary familial GIST syndrome were discussed above (Figure 1 and Figure 3).

Primary familial GIST syndrome has heritable mutations in either the *KIT* [97,98,99,100] or *PDGFRA* [101,102,103,104] genes that have been identified in several families with a predisposition to the early development of multifocal gastric and small bowel GISTs. In addition, patients with germline *KIT* mutations sometimes present with skin hyperpigmentation, dysphagia, or gastrointestinal autonomic nerve tumors, such as paragangliomas [97,98,99,100]. In contrast, hereditary *PDGFRA* mutations are associated with intestinal fibromatosis and inflammatory fibroid polyps, formerly classified as intestinal neurofibromatosis/neurofibromatosis 3b (INF/NF3b) [102,103,104].

## 9. Conclusions

In the past 20 years, the advancement of molecular genetics has provided an increasing understanding of GISTs, which has helped clinicians diagnose and treat patients with GISTs in daily practice. The diagnosis of GISTs becomes much more accurate by using IHC (CD117/DOG1) and molecular analysis (*KIT/PDGFRA*), both of which constitute the gold standard of diagnosis in GISTs (Figure 2 and Figure 3). Successful treatment in GISTs prolongs the survival of GIST patients and causes GISTs to become a chronic disease. The development of an effective treatment for GISTs resistant to imatinib/sunitinib/regorafenib and *KIT/PDGFRA*-WT GISTs will be a challenge in the near future. Monoclonal antibodies against KIT/PDGFRA might be the future of treatment in GISTs [27,28,61].

## Figures and Tables

**Figure 1 cancers-11-00679-f001:**
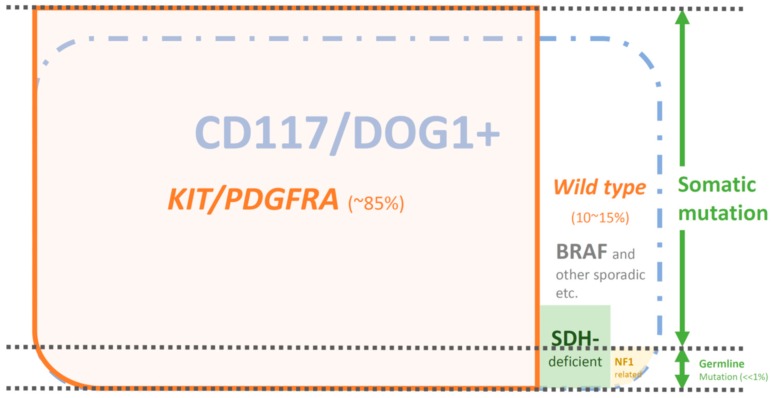
The overview of immunohistochemical staining and genetic analysis in gastrointestinal stromal tumors (GISTs). The blue dashed boundary indicates CD117/DOG1+ GISTs. The orange solid boundary indicates GISTs with *KIT/PDGFRA* mutations. The black dashed lines subgroup GISTs into somatic and germline mutations.

**Figure 2 cancers-11-00679-f002:**
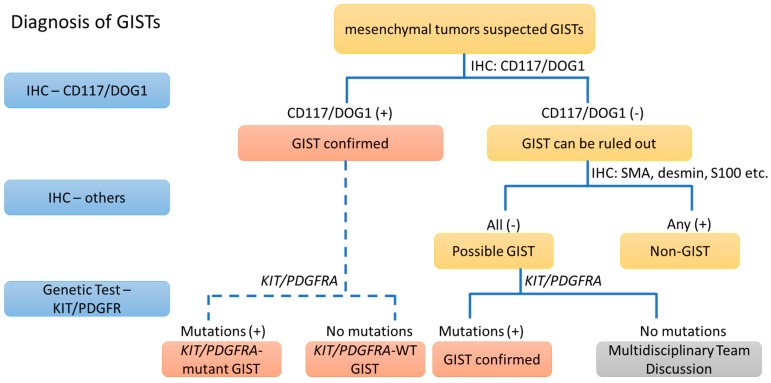
The diagnostic flow chart of GISTs by immunohistochemical staining and genetic analysis.

**Figure 3 cancers-11-00679-f003:**
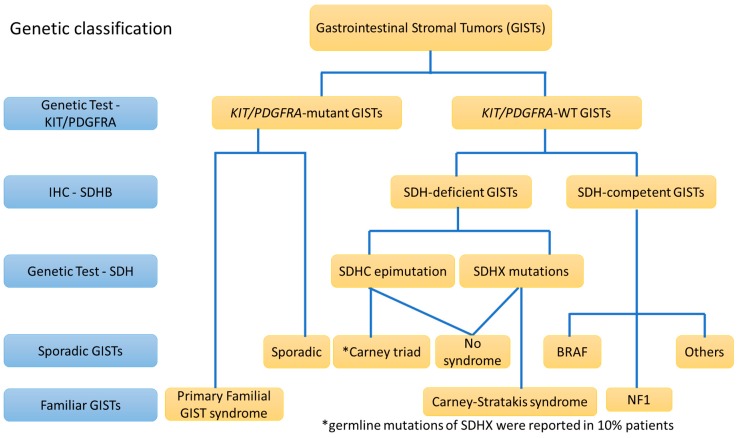
The genetic classification of GISTs. GISTs is subgrouped into *KIT/PDGFRA*-mutant and *KIT/PDGFRA*-WT GISTs. *KIT/PDGFRA*-WT GISTs can be subgrouped into succinate dehydrogenase (SDH)-deficient and SDH–competent GISTs according to SDH expression. Most of mutations are sporadic but some are germline mutations which are related to syndromic or familiar GISTs.

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
