# Peer review of "Clinical Diagnosis of Gastrointestinal Stromal Tumor (GIST): From the Molecular Genetic Point of View"

_cancers, 2019, doi:10.3390/cancers11050679_

Round 1

Reviewer 1 Report

This is a nice review article dealing with the diagnostic algorithm of GIST. On the basis of the thorough literature exploration, the authors systematically explained how to diagnose GIST. No weak point was found. However, as a reader of this article, I'd like to ask the authors to demonstrate a potential relationship between genetic findings and histological findings with appropriate photo-micrographs, such as an epithelioid subtype. 

Author Response

Dear Academic Editor:
Thanks for your comments and the point-to-point response to review 1 and reviewer 2 are described as follows:
Reviewer 1
Comments and Suggestions for Authors
This is a nice review article dealing with the diagnostic algorithm of GIST. On the basis of the thorough literature exploration, the authors systematically explained how to diagnose GIST. No weak point was found. However, as a reader of this article, I'd like to ask the authors to demonstrate a potential relationship between genetic findings and histological findings with appropriate photo-micrographs, such as an epithelioid subtype. 

Reply
Thanks for your comments. We have added some information regarding the genetics in histological subtypes in the revised manuscript.

Reviewer 2 Report

In the review manuscript, the authors comprehensively reviewed the pathogenesis and clinical views about GIST and provided their own comments on the challenge of GIST. In general, this review is well organized and clearly described, and more importantly, the authors provided accurate step-step screening for GIST diagnosis. Minor concerns need to be addressed for the revision.

1.The authors put much emphasis on the IHC/molecular diagnosis including mutation spectrum of GISTs. How the mutations drives GIST progression? The authors might add a little bit more discussion on the activated downstreams of mutated KIT/PDGFRA as the title is from molecular genetic point.
2.Any clinical trials for GIST by using monoclonal antibodies for targeting mutated KIT/PDGFRA, apart from the treatment by small molecules? It is better to add some more information about the progression of clinical treatment of GIST in the conclusion part.

Author Response

Dear Academic Editor:
Thanks for your comments and the point-to-point response to review 1 and reviewer 2 are described as follows:
Reviewer 2
Comments and Suggestions for Authors
In the review manuscript, the authors comprehensively reviewed the pathogenesis and clinical views about GIST and provided their own comments on the challenge of GIST. In general, this review is well organized and clearly described, and more importantly, the authors provided accurate step-step screening for GIST diagnosis. Minor concerns need to be addressed for the revision.
1. The authors put much emphasis on the IHC/molecular diagnosis including the mutation spectrum of GISTs. How the mutations drive GIST progression? The authors might add a little bit more discussion on the activated downstream of the mutated KIT/PDGFRA as the title is from the molecular genetic point.
Reply
Thanks for your comments. This review is about the diagnosis of GISTs so we emphasize on the molecular and genetic aspects of diagnosis rather than molecular signaling pathways in the current review to avoid confusing readers.

2. Any clinical trials for GIST by using monoclonal antibodies for targeting mutated KIT/PDGFRA, apart from the treatment by small molecules? It is better to add some more information about the progression of the clinical treatment of GIST in the conclusion part.
Reply
Thanks for your comments. We have added some information regarding monoclonal antibodies targeting KIT/PDGFRA in the revised manuscript.